# A diagnosis of the plasma waves responsible for the explosive energy release of substorm onset

N.M.E. Kalmoni [1], I.J. Rae [1], C.E.J. Watt [2], K.R. Murphy[3], M. Samara[4], R.G. Michell[4]
G. Grubbs[4] & C. Forsyth[1]

During geomagnetic substorms, stored magnetic and plasma thermal energies are explosively converted into plasma kinetic energy. This rapid reconfiguration of Earth's nightside magnetosphere is manifest in the ionosphere as an auroral display that fills the sky. Progress in understanding of how substorms are initiated is hindered by a lack of quantitative analysis of the single consistent feature of onset; the rapid brightening and structuring of the most equatorward arc in the ionosphere. Here, we exploit state-of-the-art auroral measurements to construct an observational dispersion relation of waves during substorm onset. Further, we use kinetic theory of high-beta plasma to demonstrate that the shear Alfven wave dispersion relation bears remarkable similarity to the auroral dispersion relation. In contrast to prevailing theories of substorm initiation, we demonstrate that auroral beads seen during the majority of substorm onsets are likely the signature of kinetic Alfven waves driven unstable in the high-beta magnetotail.

[1] Mullard Space Science Laboratory, University College London, Holmbury St Mary, Dorking RH5 6NT, UK. [2] Department of Meteorology, University of Reading, Reading RG6 6BB, UK. [3] Department of Astronomy, University of Maryland, College Park 20742 MD, USA. [4] NASA Goddard Space Flight Center, Greenbelt 20771 MD, USA. Correspondence and requests for materials should be addressed to N.M.E.K. (email: nadinekalmoni@gmail.com) or to I.J.R. (email: jonathan.rae@ucl.ac.uk) or to C.E.J.W. (email: c.e.watt@reading.ac.uk)

The explosive release of energy within a terrestrial substorm marks the beginning of the most dynamic and vibrant auroral display in the solar-terrestrial environment[1,2]. Stored magnetic and thermal plasma energy is quickly converted to plasma kinetic energy, resulting in dramatic changes in both the large-scale magnetic topology of the Earth's nightside magnetic field, and the increased flux of accelerated energetic particles in near-Earth space. More generally, explosive energy transfer between fields and plasma is a ubiquitous process throughout the solar system. Processes such as solar flares proceed rapidly and unpredictably, but with many common characteristics to substorms[3].

The repeatable dynamic auroral display that results from a magnetospheric substorm paints a picture of a sudden destabilisation in the nightside magnetosphere that rapidly encompasses a large area spanning many degrees of latitude and hours of local time[4]. The wide variation in magnetic field topologies and plasma populations in this coupled system means that there are an equally wide range of contrasting and conflicting theories invoked to explain the initiation of the energy release. Certainly, magnetic reconnection at the Near-Earth Neutral Line (NENL) (e.g., refs [5]), as well as fluid and kinetic plasma instabilities (e.g., ref. [6]) are all involved in the substorm process. It has been clear for decades that the majority of the stored energy is eventually released through the closure of magnetic flux via magnetic reconnection [7]. However, there is no scientific consensus on how this energy release is initiated (e.g., refs [8–11]). Optical observations of dispersive scale Alfven waves have been made in the quiet-time ionosphere (e.g., refs [12–14]).

The substorm auroral display can be approximated as a two-dimensional area in the ionosphere that maps to a huge three-dimensional magnetospheric volume of approximately $4.5 \times 10^{15}$ km$^3$. Hence, searching for the initiation mechanism in space is highly challenging and has not yet been successful. However, the Earth's magnetosphere and ionosphere form a tightly-coupled system where information is communicated over great distances by changes in electrical currents flowing along the magnetic field. Auroral displays in the ionosphere can therefore be used as an indication of the physical location in the nightside magnetosphere where stressed magnetic fields suddenly and explosively reconfigure. In the ionosphere the substorm is marked by a rapid brightening and poleward expansion of an auroral arc on the equatorward edge of the auroral oval [1,2]. Substorm auroral arcs lie on closed field lines, indicating that processes closer to Earth than the tail reconnection site are a fundamental component of the substorm. The initial signature observed prior to auroral substorm onset is the formation and evolution of auroral beads along the substorm onset arc[15–20] that are observed in >90% of events [4]. Auroral beads have also been observed simultaneously in the northern and southern hemisphere, suggesting a magnetospheric source[21].

In this paper high-cadence auroral measurements in conjunction with novel analysis techniques are used to obtain an observational dispersion relation of the plasma instability in the near-Earth plasma sheet. We compare the outcome of this analysis to kinetic wave theory in a high-beta regime to determine substorm onset aurora is strongly associated with shear Alfvén waves of short perpendicular extent. These waves accelerate electrons in the magnetosphere (e.g., ref.[22]) suggesting that the shear Alfven waves embedded in the auroral signature are not just modifying the aurora, but likely the cause of it.

## Results

**Substorm event: 18 September 2012.** On 18 September 2012, a substorm onset was observed within the field-of-view of the

MOOSE (Multi-spectral Observatory Of Sensitive EM-CCDs; e.g., ref. [23]) all-sky imagers at Poker Flat in Alaska around 09:23:00 UT. Auroral data from the 557.7 nm emission line is primarily emitted at 100–120 km altitude, characteristic of 2–10 keV electron precipitation[24]. Figure 1a shows an initial thin and faint auroral arc (indicated by the arrow in the first panel) embedded in a background of diffuse auroral precipitation. As the auroral arc begins to brighten shortly before 09:24:00 UT, periodic auroral features emerge and propagate along the arc with velocities of 1.1–1.3 km s$^{-1}$ at 110 km altitude in the eastward longitudinal direction. These auroral features are termed auroral beads (after ref. [25]) which grow in size and intensity as a function of time (Fig. 1a). The growth and evolution of these auroral beads become even clearer when an along-arc intensity profile is constructed (Fig. 1b), demonstrating that these structures propagate eastward along the onset arc.

**Auroral bead analysis.** Spatial Fourier analysis of the auroral structures at each time point provides an estimate for the perpendicular wavenumber of the beads $k_i$ in the ionosphere and indicates a superposition of multiple wave scales most evident at later times, but with a clearly identifiable preferred scale around $k_i = 1.0 \times 10^{-4}$ m$^{-1}$ (60 km ionospheric wavelength). Figure 1b confirms that, in addition to spatial structure, the auroral signal varies in time; at some longitudes, two or three periods of brightening and subsequent dimming can clearly be seen (e.g., −95°). The presence of multiple waves in the auroral data suggests the constructive interference of multiple wave modes, and hence the construction of a dispersion relation $D(\omega,k_\perp)$ is appropriate, where $\omega = \omega_r + i\gamma$, and $\omega_r$ and $\gamma$ are the real and imaginary parts of the frequency, respectively (see ref. [26], p. 10). We evaluate $\omega_r$ and $\gamma$ for each perpendicular wavenumber. The real frequency $\omega_r$ is derived from a temporal Fourier transform (as described in the Method) and $\gamma$ is estimated from the slope of any intervals of exponentially increasing power spectral density that span the time period during which beads are observed (e.g., ref. [20]). The vast majority of spatial scales show exponential growth: a classical sign of plasma instability. Using this information, we construct the dispersion relation using the real and imaginary frequency components of the instability in Fig. 2.

**Estimating the magnetospheric scales.** In order to estimate the relevant spatial scales in the near-Earth magnetosphere, we use established magnetic field modelling[27] to estimate the magnetospheric location of the auroral arc to be at a radial distance of $8.2R_E$ (Earth radii) in the magnetotail. Our ionospheric perpendicular wavenumbers are scaled up to magnetospheric perpendicular wavenumber $k_{space}$ at this radial distance, assuming that the perpendicular wavelength of the waves scales as the width of the magnetic flux tube linking the ionosphere and magnetosphere. Taking into account magnetospheric mapping, we estimate that the beads' ionospheric propagation corresponds to velocities of $v_\perp = 20-23$ km s$^{-1}$ in the plasma sheet. We discuss the relevance of this phase velocity later in this section.

**Constructing an observational dispersion relation.** Figure 2a shows the occurrence statistics of $\omega_r$ versus magnetospheric perpendicular wavenumber $k_{space}$ (see Method section for details). The relationship between perpendicular wavenumber and angular frequency is approximately linear, especially for large $k_{space}$. There is also evidence of further branches of the dispersion relation, with similar gradients but with different intercepts in angular frequency (Fig. 2a), which we discuss below in terms of solutions to the warm plasma dispersion relation. Figure 2b shows the growth rates of the

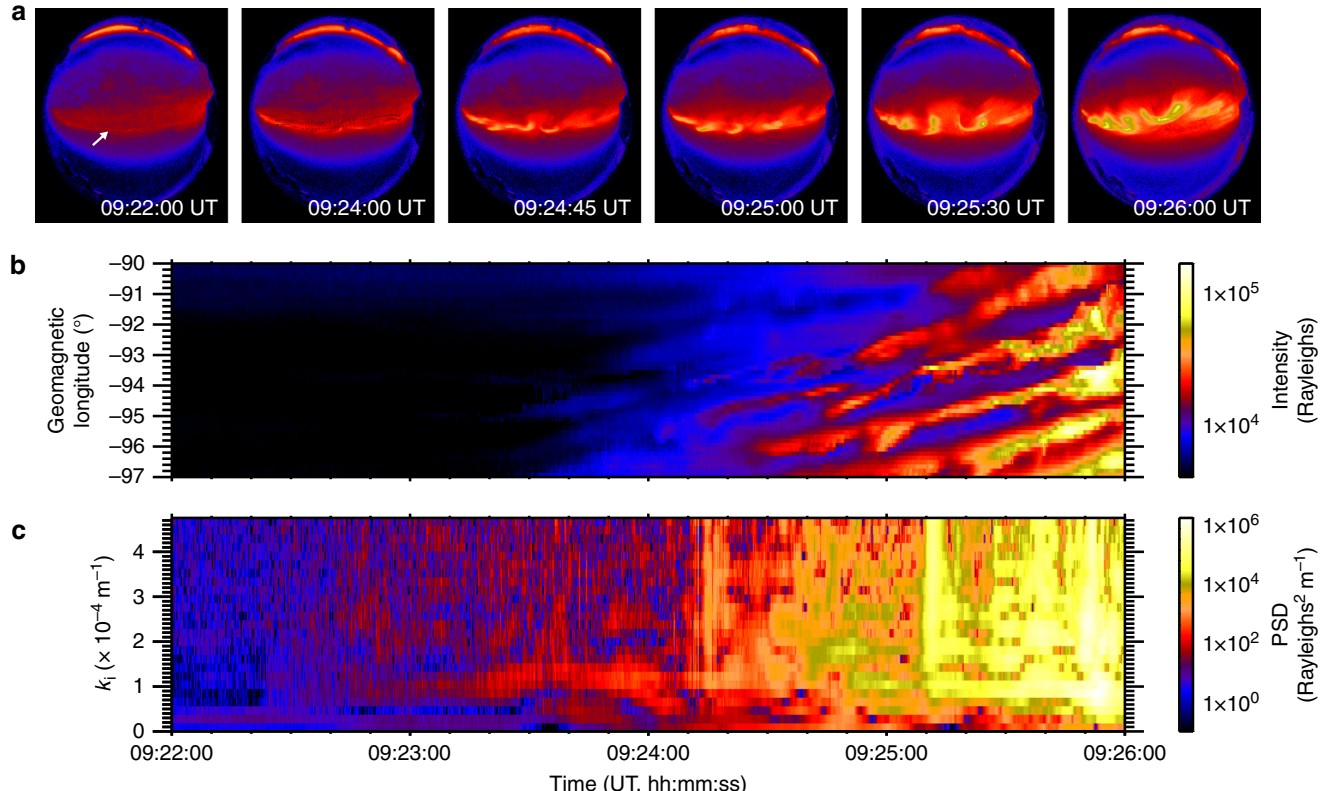

**Fig. 1** A summary of auroral observations of substorm instability. **a** Snapshots of auroral intensities as a function of latitude and longitude in the 557.7 nm wavelength through the substorm. **b** An along-arc intensity profile (keogram) as a function of geomagnetic longitude and time. **c** A spatial Fourier transform of the data shown in panel **b**

auroral beads as a function of perpendicular scale, $k_{space}$, which peak for waves with $\omega_r \sim 0.23$ rad s$^{-1}$ and $k_{space} = 9-12 \times 10^{-6}$ m$^{-1}$ and $\lambda_m = 500-670$ km ($k_i = 1.6-2.2 \times 10^{-4}$ m$^{-1}$, $\lambda_i = 29-40$ km in the ionosphere). These spatial and temporal scales can be visually identified from the variability of auroral beads shown in Fig. 1b. We note that the temporal cadence of the auroral imagers allows us to measure wave frequencies up to the Nyquist frequency of $\omega_r = 11$ rad s$^{-1}$. In Fig. 2a we only present the dispersion relation up to $\omega_r = 0.8$ rad s$^{-1}$ as there are no datapoints at these higher frequencies. From the lack of occurrence at frequencies above 0.8 rad s$^{-1}$ we can rule out that the beads in this event are caused by the Ionospheric Alfvén Resonator which has periods of $\sim$ seconds [28] as previously suggested by Sakaguchi et al.[17] and Kataoka et al.[19] Moreover, Fig. 1c shows an inverse cascade where the characteristic wavenumber of the beads decreases following substorm onset, in agreement with Lui et al.;[29] Rae et al.;[30] Kalmoni et al.,[4,20] which also supports the conclusion that the near-Earth magnetotail is the source of the instability.

**Comparison with kinetic linear dispersion relation.** To diagnose the wave mode embedded within the auroral observations, we require information on the likely magnetic fields and plasma conditions in the magnetosphere where this instability develops. Using both the magnetic field modelling (ref. [27]), and contemporaneous measurements from the nearby GOES-15 satellite, we estimate the magnetic field strength $|B| = 24$ nT at this region in the tail (see Methods). New models of plasma sheet characteristics provide estimates of the plasma sheet electron number density $n_e = 1.5 \times 10^7$ m$^{-3}$ [31], electron temperature $T_e = 2$ keV [32] and ion temperature $T_i = 3$ keV [33]. These estimates indicate that the regime in the near-Earth plasma sheet is high-beta ($\beta \sim 20$) where plasma pressure dominates over magnetic pressure. In this

region of the magnetotail, the thermal speeds ($v_{th} = \sqrt{2k_B T / m_e}$) are in far excess of the Alfvén speed ($v_A = \sqrt{B^2/\mu_0 \rho}$) and up to $v_{th}/v_A \sim 200$. Previous work has been focussed in the low-mid beta regime below $v_{th}/v_A \sim 10$ [34,35], or compared to in-situ observations where no requirement on beta was imposed (e.g., ref. [36]). A beta regime of 20, however, is far outside the bounds of current comparisons of ultra-low frequency shear Alfvén waves. We therefore extend the search for solutions of the warm dispersion relation into the extreme high-beta location recently diagnosed at the inner edge of the plasma sheet. The solutions of the warm plasma dispersion relation are shown together with the observational counterpart in Fig. 2a.

The dashed line in Fig. 2a indicates the real frequency solutions of the warm plasma dispersion relation in an infinite homogenous plasma using the above parameters and for a parallel wavelength of $\lambda_\| = 1.8 R_E$ (the dot-dashed lines give solutions for wavenumbers ± 10%). Observations of the parallel wavenumber are, in practice, the most challenging free parameter to measure in a plasma, and we are unable to constrain this parameter from the available observations. Hence, $k_\|$ is a free parameter which is determined from the full solution of the dispersion relation (Fig. 7) to be both realistic and to agree best with our observations.

The solutions show remarkable agreement with the auroral dispersion relation and diagnose this mode as a shear Alfvén wave with short perpendicular wavelength ($k_\perp \gg k_\|$)[37–39]. In the regime $v_{th} \gg v_A$, this wave is often referred to as a kinetic Alfvén wave. Additional dispersion curves may be evident in Fig. 2a with a similar gradient but different intercepts in angular frequency, which we can attribute to a more broadband source of shear Alfvén waves with a range of parallel wavenumbers ($k_\|$ corresponding to 1.8$R_E$; See Fig. 7). However, Fig. 2 shows that the

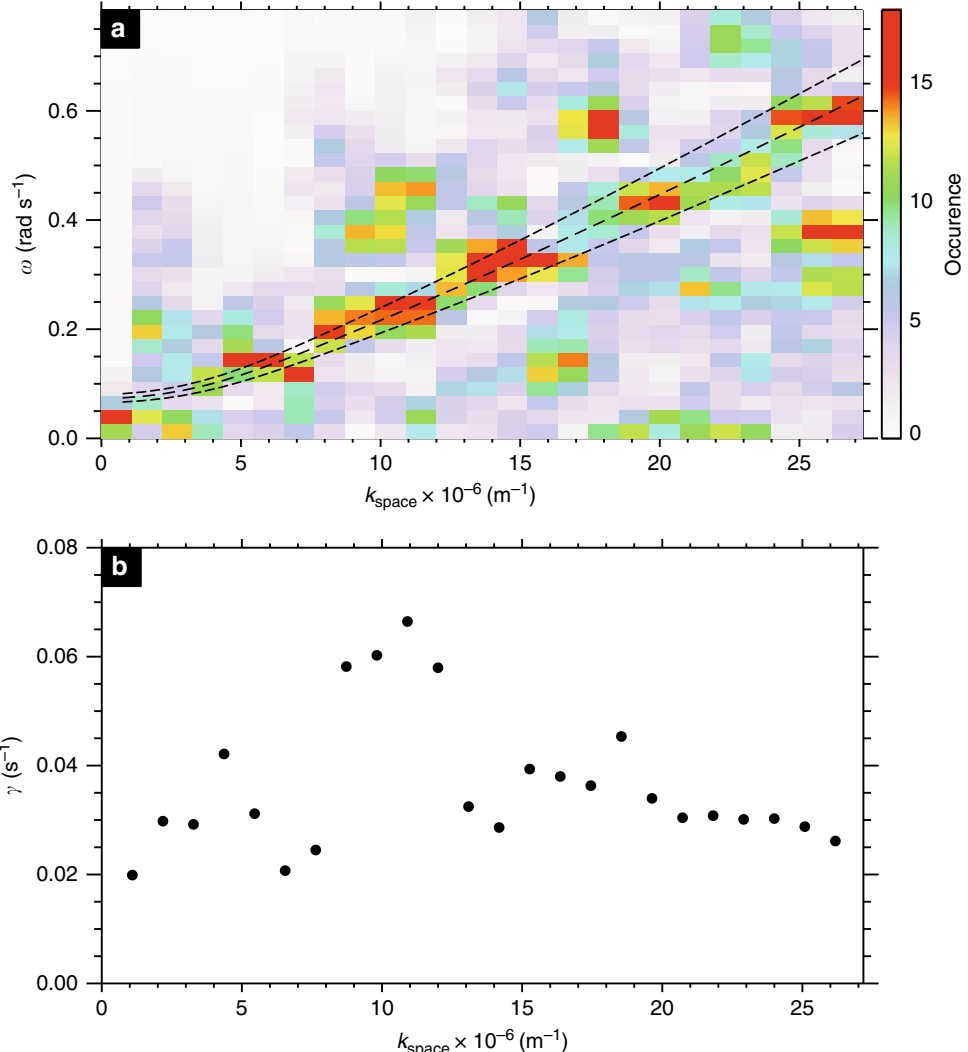

**Fig. 2** A comparison of the observational dispersion relation with solutions to the warm plasma dispersion relation in the near-Earth magnetotail using realistic plasma parameters. **a** Observational dispersion relation shown in colour, along with analytical solutions to the full warm plasma dispersion relation for an infinite uniform plasma (dashed lines—central dashed line shows solutions for a parallel wavelength of approximately $1.8R_E$, and the dashed-dot lines indicate parallel wavelengths of $1.8R_E \pm 10\%$). A sensitivity analysis of the solutions to the warm plasma dispersion relation with respect to varying the magnetic field, number density and temperature can be found in Method. **b** Observational growth rates (imaginary part of the frequency) as a function of wavenumber that demonstrate that these wavenumbers are growing exponentially, as would be expected in the linear stage of an instability

dominant $k_\perp \sim 10 \times 10^{-6}$ m$^{-1}$ shown in Fig. 1c lies along the dispersion curve shown in Fig. 2a, with $\omega \sim 0.25$ rads$^{-1}$.

**Auroral bead perpendicular phase velocity**. We can use the solutions of the full warm plasma dispersion relation to calculate a theoretical perpendicular velocity of the waves, given by $v_\perp = \frac{\omega_r}{k_\perp}$. Figure 3 shows the remarkable agreement between the phase velocities obtained from the solutions of the warm plasma dispersion relation for a wave with $\lambda_\parallel = 1.8R_E$ and the bead along-arc propagation velocities determined from observations from $k_\perp \sim 5 \times 10^{-6}$ m$^{-1}$, with the dominant auroral bead scales being significantly higher at $k_\perp \sim 10 \times 10^{-6}$ m$^{-1}$. Note that our observational analysis window follows the latitudinal centroid of the substorm onset arc (see Methods) and therefore moves poleward during the interval as the substorm onset arc also moves poleward. This accounts for any apparent phase motion arising from plasma injected into the instability region and/or dipolarisation of the nightside field in the radial direction. Hence, we are

confident that our analysis will obtain the true azimuthal phase velocity of the auroral features.

We reiterate that we have chosen a parallel wavelength of $1.8R_E$ in order to match our observational results. Although the results of the warm plasma dispersion relation are relatively insensitive to the choice of $k_\parallel$, in the real magnetosphere, this wavelength may not be constant along the geomagnetic field line, particularly when travelling along the field into regions of increasing Alfven speed (e.g., ref. [38]). Nevertheless, using a constant parallel wavelength, we find excellent agreement with our observational results. Moreover, since our perpendicular phase velocity estimates are also constant, we believe that a choice of a constant parallel wavelength is appropriate for this interval.

## Discussion
The methods and dataset utilised in this study allow us to characterise both the temporal (real and imaginary frequency components) and spatial scales (wavenumber) from auroral data.

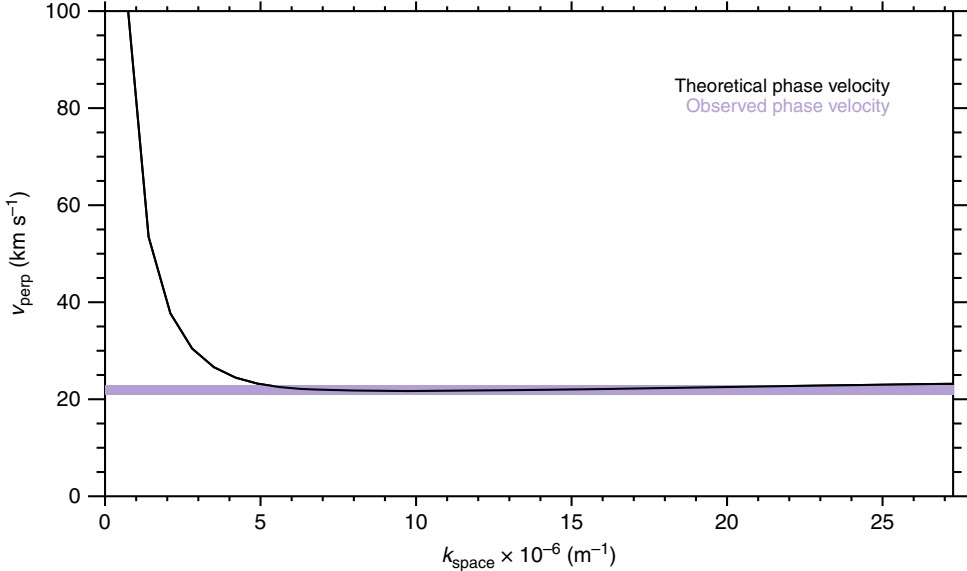

**Fig. 3** Comparison of observed and theoretical phase velocities. Perpendicular phase velocity calculated from the theoretical dispersion relation (central dashed line Fig. 2a) indicated in black, with the measured bead perpendicular propagation velocities in the ionosphere which have been mapped to the plasma sheet indicated by the purple shaded region

Previously, only the wavenumber and imaginary frequency have been resolved (e.g., ref. [20,30]) through this type of optical Fourier analysis (due to a lower Nyquist frequency and Nyquist wave-number). The increased spatial resolution and, critically, an order or magnitude higher temporal cadence available from MOOSE allows us to identify the crucially important real frequencies, and construct the first observational dispersion relation of substorm onset without introducing any aliasing effects.

In previous studies, numerous instabilities have been proposed to play a fundamental role in the physics of substorm onset (see ref. [6] for a review). To our knowledge, there is no dispersion relation in the literature that is able to reproduce our observed growth rates and spatial scales and the characteristic linear relationship between angular frequency and spatial scales.

Based on the instability growth rates and spatial structuring characteristics, Kalmoni et al.[20] deduced that the most likely instabilities for auroral beads were the Cross-Field Current Instability (e.g., [6,40]) and some form of Ballooning instability (e.g., refs [41–43]). With the addition of angular frequencies, we are now able to identify that neither instability could be responsible for the azimuthal structuring of the substorm onset arc. Instead, we apply the current theory of shear Alfvén waves with short perpendicular scales (or kinetic Alfvén waves) to this high-beta regime, and find that the observed dispersion relation for kinetic Alfvén waves in a realistic plasma environment reproduces our observational results exceptionally well.

Indeed, kinetic Alfven waves at substorm onset have previously been identified from in-situ measurements[44] and kinetic Alfven waves have been observed in the equatorial magnetotail in this region closely conjugate to auroral brightenings[45–47]. Note that we have no low-altitude evidence of the type of electron acceleration mechanism that causes this type of aurora. Motoba and Hirahara[48] showed evidence from low-altitude satellite measurements where electrons at a range of pitch angles are observed at the same time as the onset arc. Previous evidence from FAST spacecraft measurements and kinetic simulations of Alfvenic acceleration[49] indicates that acceleration due to large Alfven waves can result in mono-energetic populations of electrons with a wide range of pitch angles (Fig. 2)[49] and confirmed by the

simulations of Watt and Rankin[38]. Both the in-situ and ground-based results presented in this paper point to an Alfven wave instability operating at substorm onset and providing an acceleration mechanism for the onset aurora. We further note here that the solutions to the warm plasma dispersion relation do not reproduce the imaginary part of the frequency shown in Fig. 2b; these analytical solutions show only damped modes. However, the input to the warm plasma dispersion relation lacks any free energy source to support growing waves and hence would not sustain any instability. In reality, the magnetotail is likely to support pressure gradients[50–52], shear flows[41,53] and electron anisotropies (e.g., ref. [32]), all of which may result in growing frequency solutions (frequencies with positive imaginary parts). Viñas and Madden[54] in particular suggest that a shear flow-ballooning instability operating in the magnetosphere would create unstable shear Alfvén waves. Our observational and theoretical results suggest an urgent need to revisit previous analyses of plasma instabilities in inhomogeneous plasma given more recent knowledge of possible free energy sources and the determination of the cause of the auroral substorm presented in this work.

Indeed, previous work by Watt and Rankin[38] demonstrates that Alfvén waves with short perpendicular scales launched from the warm magnetotail can accelerate electrons to form visible aurora remarkably efficiently, providing the key missing link in explaining the magnetospheric source of auroral beading that marks substorm onset in the ionosphere.

These results drive a search for conjugate in-situ observations that can measure the local plasma parameters in the location of the instability and determine the source of the free energy that drives the instability.

We use state-of-the-art auroral measurements to calculate the first observational dispersion relation for the substorm onset instability, and find that there is a clear relationship between the temporal and spatial characteristics of the auroral display. We can now determine the relationship between the frequency and wavelength of the substorm instability to test against existing theories. However, previous theoretical treatments of plasma instabilities[55], do not show these same characteristics. We apply

the current theory of kinetic Alfvén waves into the high-beta regime of the near-Earth warm plasma sheet and discover that driven kinetic Alfvén waves reproduce the temporal and spatial characteristics of the substorm instability with no other requirements. Hence, we can diagnose which waves exist in the magnetosphere at substorm onset and determine the driver of the explosive energy release in the magnetospheric substorm.

Ever since it's discovery in 1964[1], the processes that cause the substorm onset aurora to explosively brighten have been causally linked to the explosive energy release from nightside magnetic reconnection processes. Energy stored in the stretched magnetotail and plasma environment is suddenly and, seemingly unpredictably, released in the form of energised particle

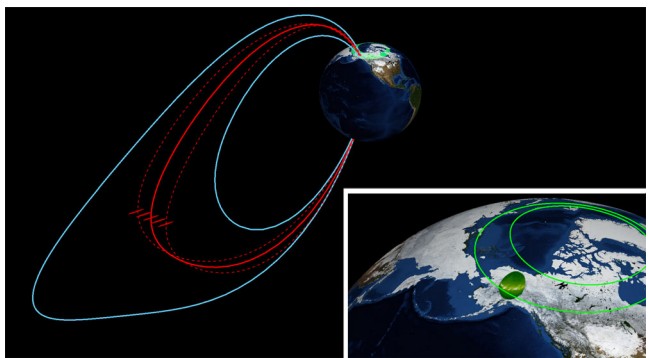

**Fig. 4** To-scale estimate of the spatial characteristics of the substorm instability. Auroral observations are mapped to the equatorial magnetotail using a T96 magnetic field model[27]. The auroral beads map to the transition region between dipolar and stretched magnetic field topology. Inset: Auroral bead measurements used to estimate the spatial scales of the substorm onset region in space, with the inner and outer bounds of a Feldstein auroral oval for reference[62]

populations, enhanced current systems[11] and enhancements of the Van Allen Radiation belts (e.g.,[56–58]). The process that initiates the auroral substorm, however, lies deep in a closed field line region where reconnection is not typically thought to occur (Fig. 4). The question then, is what physical process drives the destabilisation of the substorm onset arc?

Our results demonstrate that the auroral beads observed along the substorm onset arc are the ionospheric manifestation of a shear Alfvén wave instability in a high-beta region of the plasma sheet operating near the transition between stretched and dipolar magnetic field topology, as schematically shown in Fig. 3. This conclusion inspires the immediate search for the location and characteristics of the instability that marks the start of the magnetospheric substorm, significantly moving forward a problem that has eluded the space science community since its discovery in 1964.

## Methods

**Substorm identification and auroral camera data**. We present a substorm event on the 18 September 2012 that is marked by sudden brightening and formation of beads along the substorm onset arc, which eventually lead to auroral breakup and poleward expansion. Traditional analysis centres on the analysis of a North-South slice (keogram) through the centre of the initial auroral brightening to demonstrate the brightening and poleward expansion of the aurora denoting auroral substorm onset[1,2], and the conjugate ground magnetometer measurements of the associated current systems[59]. Figure 5 shows an overview of the event. The substorm onset arc can be identified from the keogram (Fig. 5a) at 65.75° geomagnetic latitude. The onset arc visibly brightens at 09:24:00 UT and shortly after breaks up. The auroral display advances poleward to 67.5° geomagnetic latitude and the aurora remains active for at least 10 min.

The auroral substorm is also accompanied by a geomagnetic bay of ~ 120 nT in the H-component of the magnetic field measured by the magnetometer at Poker Flat (See Fig. 5b). The formation of the bay starts ~ 09:22:00 UT and reaches its lowest point at −50 nT at 09:28:30 UT. This is the first substorm in a sequence of multiple substorms with successive H-bays of around 100–150 nT forming and the overall H-magnetic field dropping to −245 nT at 10:25:00 UT.

In our analysis we use data from the 557.7 nm wavelength auroral emission observed by the MOOSE imagers. The increased temporal cadence (3 Hz) and

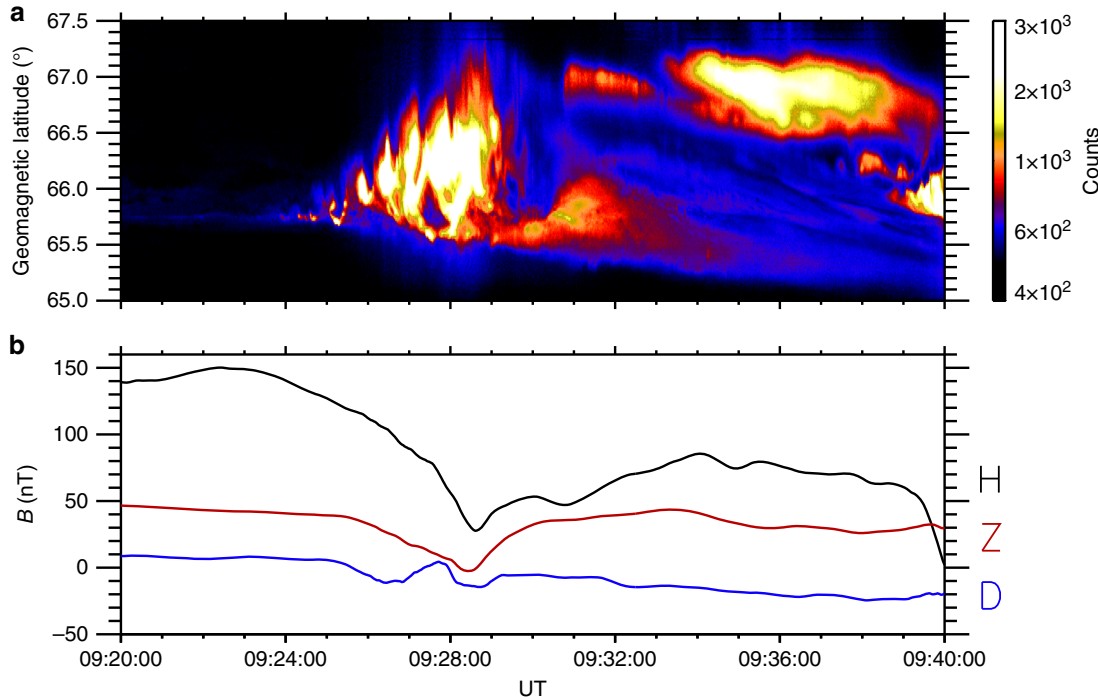

**Fig. 5** An overview of the auroral and magnetic measurements during the substorm event. **a** Vertical slice through centre of the ASI to generate North-South keogram of event. Faint pre-existing arc is visible from 09:20:00 UT at 65.75 MLAT. The arc visibly brightens at 09:24:00 and shortly after the aurora expand poleward to 67.2 MLAT. **b** The deflections in the magnetic field components at the POKR magnetometer show the formation of the magnetic bay around the same time as the auroral signature of the substorm

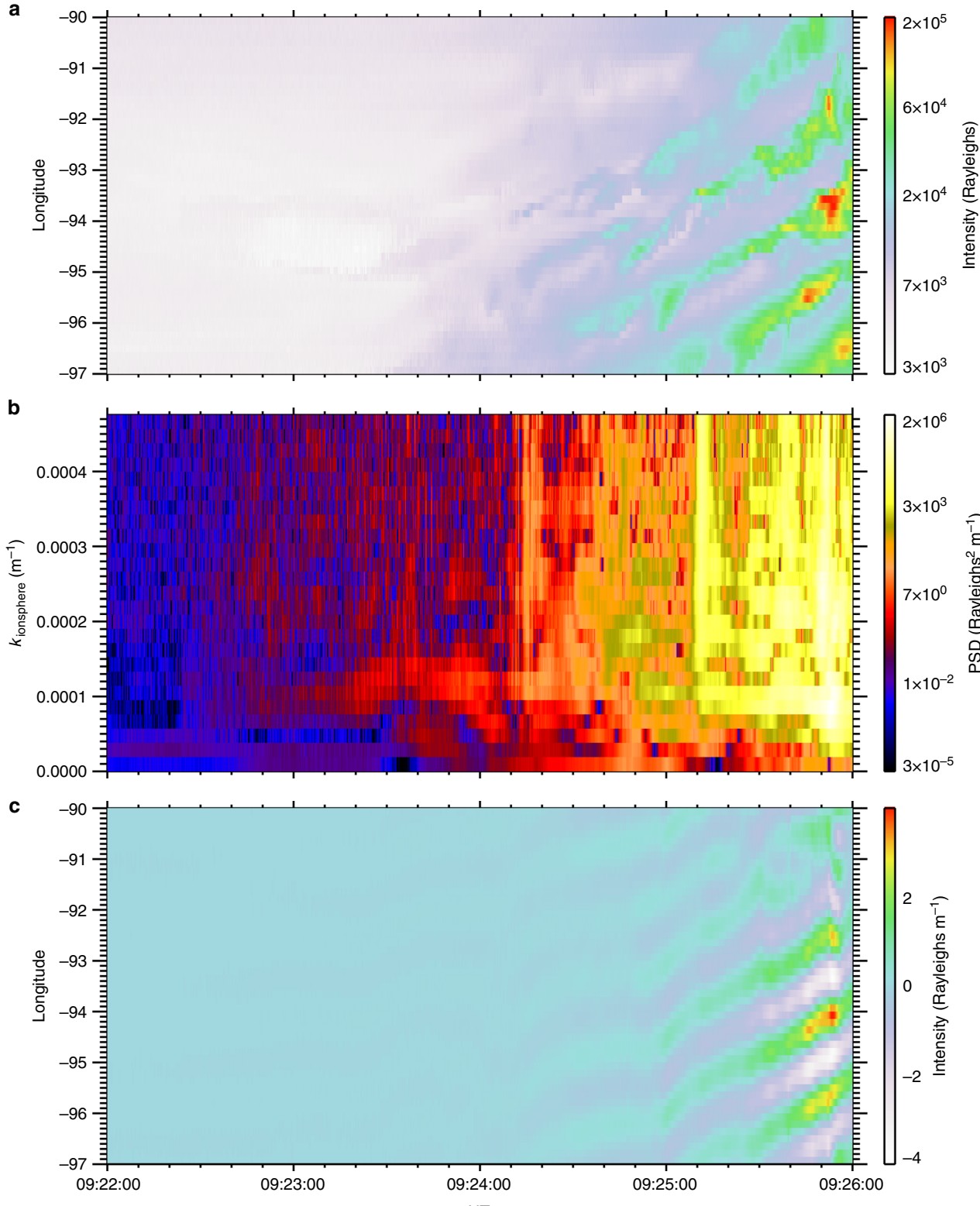

**Fig. 6** A demonstration of the algorithm to estimate frequency as a function of wavenumber. **a** The original along-arc keogram showing auroral beads propagating along the substorm onset arc in the Eastward direction and **b** their dynamic power spectral density calculated from a spatial Fourier transform of panel a in the longitudinal direction. **c** An example wavenumber of $k = 0.8 \times 10^{-4}\,\mathrm{m}^{-1}$ (white vertical dashed line in panel **b**) as illustration of our methodology. Only information for an individual spatial scale is used to inverse Fourier transform to the spatial domain. This so-called 'filtered' keogram is then used to find the real frequency component as a function of wavenumber in order to construct the observational dispersion relation

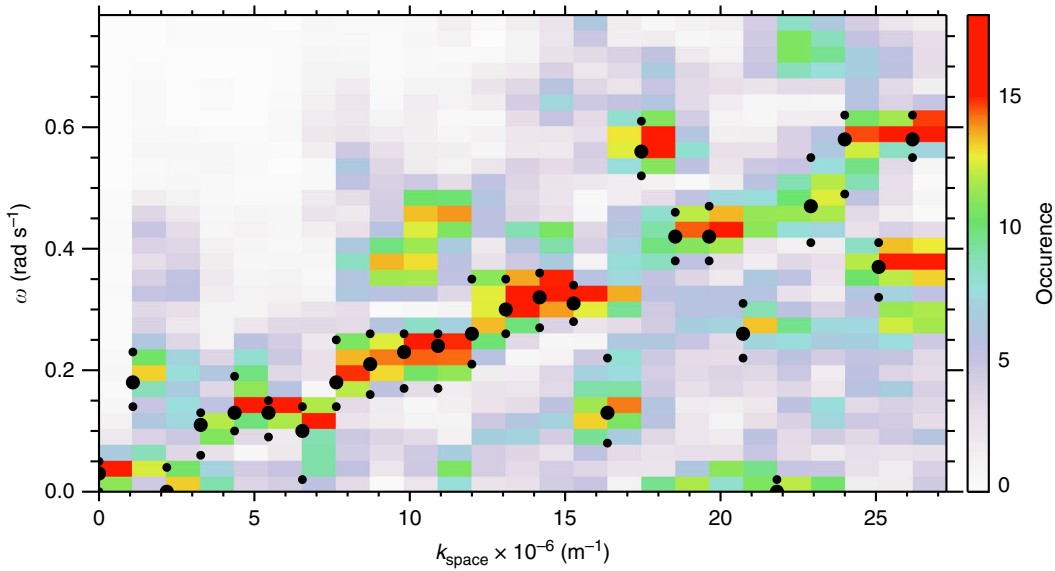

**Fig. 7** Observational dispersion relation. The peak occurrences (big dots) and the Full-Width Half Maximum (small dots) are indicated for each wavenumber. For the majority of spatial scales, the peak occurrences lie along the same line. At some spatial scales, multiple peaks make up the distribution in omega. Only the highest peaks in the distribution are shown by the dots

spatial resolution (500 m at zenith) of MOOSE, in comparison to other all-sky imagers, such as the white-light THEMIS All-Sky Imagers (ASIs) [60] allows the aurora to be significantly better characterised in this study. The data were calibrated using techniques outlined in Grubbs et al.[23]

**Construction of an observational dispersion relation.** The underlying azimuthal structuring of the auroral arc is quantitatively characterised by use of an arc-tracking algorithm to determine the latitudinal position of the arc in the field of view of the ASI as the substorm develops at each location between −97.0 and −90.0 geomagnetic longitude prior to the arc breaking-up into multiple auroral arcs, at 09:26:00 UT. By tracking the arc for a significant amount of time, we can generate an along-arc keogram (Fig. 1b) which contains both temporal and spatial information of the auroral arc. In producing the along-arc keogram, it is assumed that the 557.7 nm auroral emission is located at 110 km altitude and has very little vertical extent. This assumption does not affect the projection of the beads at zenith of the imager, but could have some affects if the assumption does not hold towards the edges of the auroral arc observed. In this event, the maximum extent of the arc away from zenith is 60°. This could introduce a maximum uncertainty of 15−20 km on our ~ 100 km bead spatial scales assuming that the 557.7 nm emission can extend between 100–120 km altitude. However, since we are applying a windowing function to our measurements to perform subsequent Fourier analysis, the error on these measurements is much smaller as the signature of the beads is weighted towards the observations towards the centre of the field of view.

Hence, we are able to have sufficient temporal and spatial information with which to determine both the angular frequency $\omega_r$ and the perpendicular scale $k_\perp$ of the auroral arc and construct an observational dispersion relation of the substorm onset arc. This is done by the following method:

1. A Fast Fourier Transform (FFT) is used in the longitudinal direction to decompose the signal into the individual wavenumber components that make up the beads (Fig. 6a, b). The resulting power spectral density can be used to decompose the signal into individual wavenumbers.
2. A reverse FFT is used to transform the signal for each wavenumber back into the time domain to construct a filtered along-arc keogram i.e., a signal which only contains a single wavenumber component of the beads. Figure 6c shows the band-pass filtered keogram for $k_i = 0.8 \times 10^{-4} \, m^{-1}$.
3. Temporal frequency analysis of individual horizontal slices of the filtered keogram is performed to measure the real frequency $\omega_r$ individually for all longitudes ranging from −94.6° to −92.2°. The temporal frequency signal for each individual wavenumber is normalised, as the lower wavenumbers constitute a larger component of the total signal of the beads.
4. Each resultant analysis is used for input into Fig. 2 to create the observational dispersion relation $D(\omega_r, k_\perp)$.

This analysis results in the observational dispersion relation for the real frequency component, omega as a function of wavenumber. Figure 7 shows the observational dispersion relation, which is the same as shown in Fig. 2a, with the peak occurrences for each wavenumber indicated by larger circles, and the Full-Width Half Maximum of the peaks indicated by the smaller circles. Figure 7 shows

that for the majority the peaks very clearly follow the line, with some exceptions, particularly at higher wavenumbers.

**Application of current theory into the warm plasma regime.** We follow the method of Horne[61] to obtain the full warm plasma dispersion relation. This has the general form:

$$D(k, \omega) = An^4 + Bn^2 + C \qquad (1)$$

where $n = ck/\omega$ is the refractive index and $A$, $B$, and $C$ are given by:

$$A = \epsilon_{xx} \sin^2 \psi + 2\epsilon_{xz} \cos \psi \sin \psi + \epsilon_{zz} \cos^2 \psi \qquad (2)$$

$$B = -\left[ \left( \epsilon_{xy} \sin \psi - \epsilon_{yz} \cos \psi \right)^2 + \epsilon_{xx}\epsilon_{zz} - \epsilon_{xz}^2 + A\epsilon_{yy} \right] \qquad (3)$$

$$C = \left( \epsilon_{xx}\epsilon_{zz} - \epsilon_{yz}^2 \right)\epsilon_{yy} + \left( \epsilon_{xy}\epsilon_{zz} + 2\epsilon_{xz}\epsilon_{yz} \right)\epsilon_{xy} + \epsilon_{xx}\epsilon_{yz}^2 \qquad (4)$$

The angle $\psi$ is the angle between the wavenormal and the magnetic field, and the $\epsilon_{ij}$ are the elements of the dielectric tensor:

$$\epsilon(k, \omega) = \left( 1 - \frac{\omega_p^2}{\omega^2} \right)I + \sum_s \frac{\omega_{ps}^2}{\omega^2} \sum_{m=-\infty}^{\infty} \int_0^\infty 2\pi v_\perp dv_\perp \int_{-\infty}^{\infty} dv_\parallel \left[ \frac{\frac{m\Omega_s}{v_\perp}\frac{\partial f}{\partial v_\perp} + k_\parallel \frac{\partial f}{\partial v_\parallel}}{\left( \omega - m\Omega_s - k_\parallel v_\parallel \right)} \right] \Pi \qquad (5)$$

where $s$ denotes plasma species, $\omega_{ps}^2 = n_s q_s^2/(\varepsilon_0 m_s)$ is the plasma frequency of species $s$, $\omega_p = \sum_s \omega_{ps}$, $k_\perp$ and $k_\parallel$ are the perpendicular and parallel wavenumbers, $v_\perp$ and $v_\parallel$ are the perpendicular and parallel velocities, $\omega$ is the wave frequency, and $\Omega_s$ is the gyrofrequency of species $s$. The $\Pi$ tensor is given by:

$$\Pi = \begin{bmatrix} \frac{m^2\Omega_s^2}{k_\perp^2}J_m^2 & iv_\perp \frac{m\Omega_s}{k_\perp}J_m J'_m & v_\parallel \frac{m\Omega_s}{k_\perp}J_m^2 \\ -iv_\perp \frac{m\Omega_s}{k_\perp}J_m J'_m & v_\perp^2 \left(J'_m\right)^2 & -iv_\perp v_\parallel J_m J'_m \\ v_\parallel \frac{m\Omega_s}{k_\perp}J_m^2 & iv_\perp v_\parallel J_m J'_m & v_\parallel^2 J_m^2 \end{bmatrix} \qquad (6)$$

where the argument of the Bessel functions $J_m$ is $\mu = k_\perp v_\perp/\Omega_s$. We assume that the distribution functions of both electrons and protons in the warm plasma in the plasma sheet are Maxwellian.

Note that since all parameters are assumed homogeneous, these solutions are most applicable to a plasma where the wavelengths are small compared to the gradient scale lengths. Theoretical treatments are derived from a planar wave assumption. These assumptions may not be strictly valid in the magnetotail, however this method provides important insight into the real part of the dispersion relation[34] and therefore allows us to diagnose the wave mode present. The imaginary part of the temporal frequency is highly sensitive to the presence or absence of free energy and so motivates the future development of numerical

treatments of the warm plasma dispersion relation in inhomogeneous plasma to capture the source of free energy that drives the instability.

Solutions are obtained using a Newton–Raphson algorithm for complex frequency given real wavevector, $\mathbf{k} = (k_\perp, k_\parallel)$. Other branches were investigated but no other solutions at frequencies $0.1 < \omega_r < 1.5$ rads$^{-1}$ were found. Solutions for nearby $k_\perp$, $k_\parallel$ are shown in Fig. 8.

The results from the theoretical dispersion relation have been indicated in Fig. 2 for a parallel wavelength of $1.8R_E$. We use this theoretical calculation, together with the occurrence peaks from the dispersion relation (represented by the large circles in Fig. 7) to perform a Chi-squared test between the observational and expected results. We find that the Chi-squared value is 1.15, and is statistically significant to beyond the 99.99% level, indicating that the observations are very close to the expected values for a kinetic Alfven wave with these characteristics (Fig. 9).

**Sensitivity analysis**. Solutions depend on four plasma parameters: magnetic field strength, number density, electron temperature and ion temperature (assuming the plasma is made up of electrons and protons only), and one wave parameter (parallel wavenumber).

In each panel of Fig. 10 the solid line shows how the frequency varies with perpendicular wavenumber for constant $k_\parallel = 5.4 \times 10^{-7}$ m$^{-1}$, $|B| = 24$ nT, $n = 1.5 \times 10^7$ m$^{-3}$, $T = 2$ keV and $T_i = 3$ keV (i.e., the solutions shown by the dashed line in Fig. 2a).

The dashed lines and dot-dash lines show solutions when the indicated parameter is varied by 50% above and below the quoted value respectively.

There are some small variations in the solutions with $n$, $T_e$ and $T_i$. If we had performed the same analysis with different plasma parameters (up to ± 50%, we would have been able to match our observational dispersion relation shown in Fig. 2a with slightly different $k_\parallel$. For example, solutions for $n = 7.5 \times 10^6$ m$^{-3}$ and

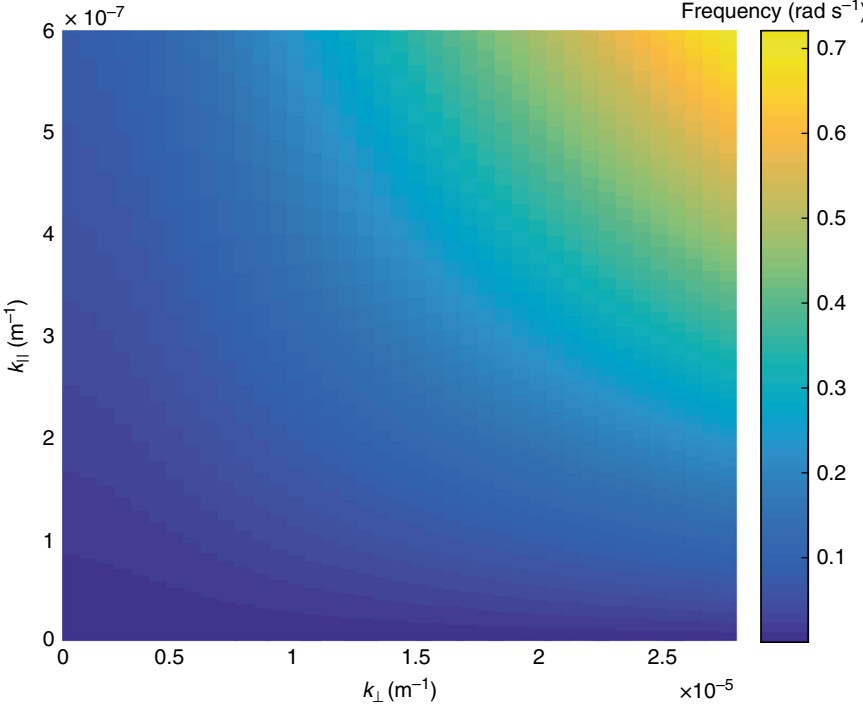

**Fig. 8** Solution to the warm plasma dispersion relation. Frequency is shown as a function of parallel wavenumber as a function of perpendicular wavenumber and angular frequency

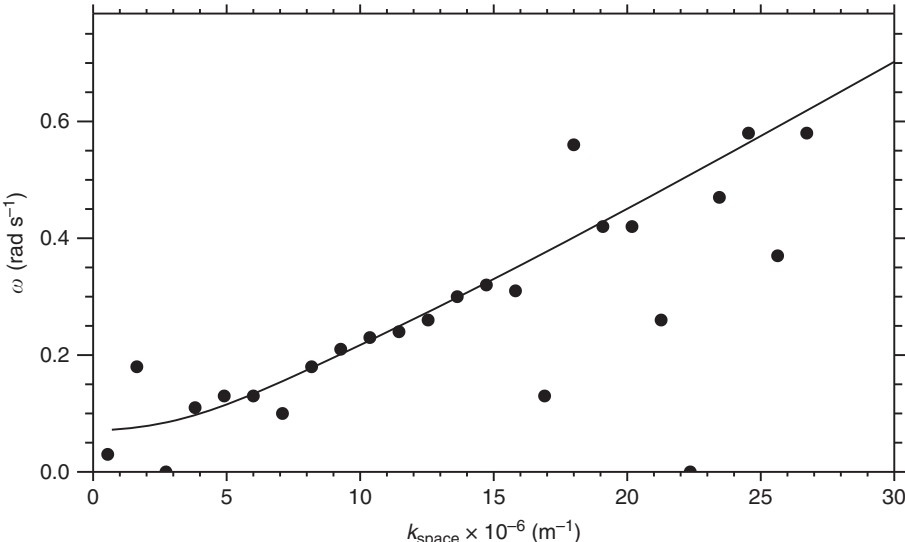

**Fig. 9** Comparison of observed and theoretical dispersion relations. Results from the theoretical dispersion relation for a parallel wavelength of $1.8R_E$ are shown, together with the peaks for each wavenumber determined from the observational dispersion relation used to conduct the Chi-squared test

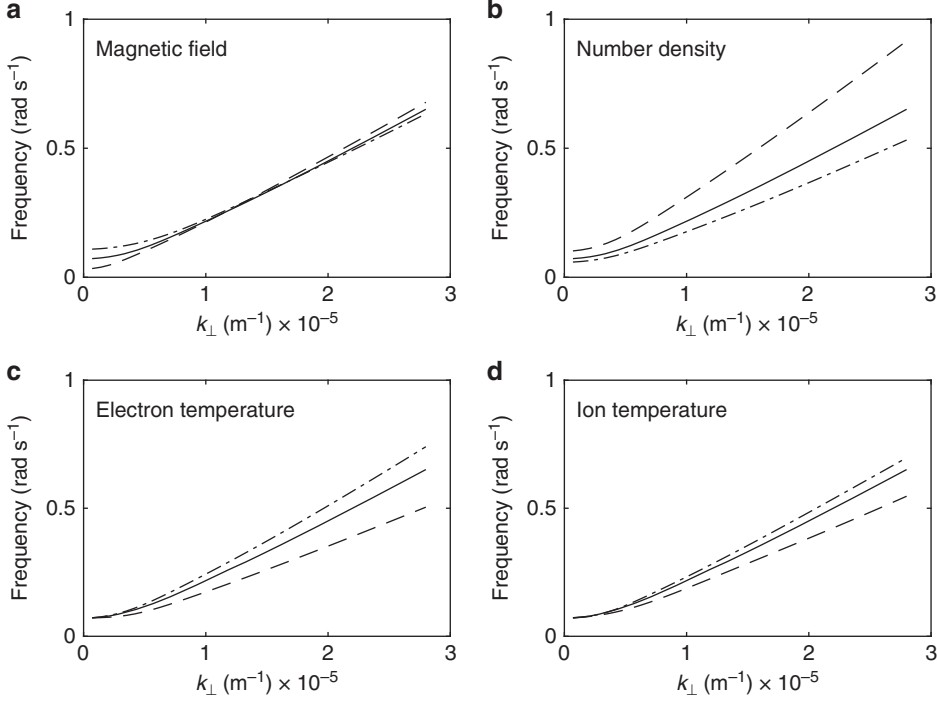

**Fig. 10** Solutions of the warm plasma dispersion relation with varying parameters to assess the sensitivity of the solutions. The sensitivity is assessed based on varying each parameter by ± 50%. **a** $k_{||}$, $n$, $T_e$ and $T_i$ are all kept constant and $|B|$ is varied from 12 nT (dashed line) to 36 nT (dot-dash line). Varying the magnetic field strength does not impact the solution strongly. **b** $k_{||}$, $B$, $T_e$ and $T_i$ are all kept constant and $n$ is varied from $7.5 \times 10^6\,\mathrm{m}^{-3}$ (dashed line) to $2.25 \times 10^7\,\mathrm{m}^{-3}$ (dot-dash line). **c** $k_{||}$, $B$, $n$, and $T_i$ are kept constant and $T_e$ is varied from 1.0 keV (dashed line) to 3.0 keV (dot-dash line). **d** $k_{||}$, $B$, $n$, and $T_e$ are kept constant and $T_i$ is varied from 1.5 keV (dashed line) to 4.5 keV (dot-dash line)

$k_{||} = 3.5 \times 10^{-7}\,\mathrm{m}^{-1}$ (or a wavelength of $2.8 R_E$) match our observational dispersion relation just as well as the solutions shown in Fig. 2a.

**Code availability**. Computer code is available from the corresponding authors upon request.

## Data availability
Calibrated 557.7 nm auroral data is available via FigShare and can be accessed at https://doi.org/10.6084/m9.figshare.6960122.v1

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

## Acknowledgements

N.M.E.K. was supported by STFC studentship ST/K50239X/1, and by MSSL's Consolidated Grant ST/N000722/1. While working on this paper N.M.E.K. visited co-authors at NASA GSFC. This trip was funded by NNH11ZDA001N-GEO 11-LCAS11-0023 and the E.A. Milne travelling fellowship from the Royal Astronomical Society, London. I.J.R. is supported in part by ST/N000722/1 and by NERC grant NE/L007495/1. C.E.J.W. is supported in part by ST/M000885/1 and ST/R000921/1. K.R.M. was supported by NSF Grant Number 1602403. R.M. was supported by NSF grants AGS1456161 and AGS1456129 and NASA grant NNX15AG06G. M.S. and G.G. were supported by NASA ROSES funding through NNH13ZDA001N-HTIDS 13-HTIDS13_2-0024 and NNH11ZDA001N-GEO 11-LCAS11-0023. C.F. is supported by a NERC Independent Research Fellowship NE/N014480/1.

## Author contributions

N.M.E.K., I.J.R. and C.E.J.W. wrote the manuscript. N.M.E.K. performed auroral analysis. I.J.R. provided guidance and discussion for analysis. C.E.J.W. found solutions of warm plasma dispersion relation. K.R.M. provided code & guidance for auroral analysis. M.S. provided useful discussion and information on instrument limitations. R.G.M. provided useful discussion, instrument expertise and star maps of imagers. G.G. calibrated auroral data. C.F. produced 3D schematic.

## Additional information

**Competing interests:** The authors declare no competing interests.

