## [Peer Review File · Nature Communications]

Reviewers' comments:

Reviewer #1 (Remarks to the Author):

The onset is a major issue of auroral substorms/magnospheric substorms and has been hotly debated. The paper deals with one of the precursors called "beads" which was discovered some years ago. The paper consists of an excellent data set and a good insight into details of theoretical analysis of the waves associated with this phenomenon. Therefore, I am happy to recommend its publication in Nature.

Reviewer #2 (Remarks to the Author):

This paper has attempted to construct the dispersion relation of the auroral structuring to diagnose the waves responsible for the explosive energy release from substorm onset, using high-resolution auroral imagery data. The authors have examined a particular case of auroral beads developing along an onset arc on 18 September 2012 and demonstrated that the azimuthal structuring of the arc is the manifestation of kinetic Alfvén waves driven in the high-beta regime of the near-Earth magnetotail, not any of previously proposed instabilities (such as, cross-field current instability, ballooning instability). This may have potentially important implications towards the understanding of how these features relate to magnetospheric activity and the importance and relevance to magnetospheric dynamics around substorm onset. However, I cannot recommend that this manuscript is published in the current form due to the lack of novel dataset/approach. Below are specific comments.

(1) The dataset and approach to determine the properties of the azimuthal structure of auroral beads are NOT novel, which is a major weakness of this study. Indeed, Sakaguchi et al. ("Azimuthal structures of ray auroras at the beginning of auroral substorms", *Geophys. Res. Lett.*, 2009) and Kataoka et al. ("Turbulent microstructures and formation of folds in auroral breakup arc", *J. Geophys. Res.*, 2011) have already estimated the growth rates and spatial structuring characteristics of onset arc based on similar high temporal/spatial resolution auroral image data and approach.

(2) The interpretation of the presented results remains largely speculative due to the lack of direct evidence from in situ measurements. The dispersion relation extracted from the aurora image data generally fits the frequency solutions expected from the theoretical approach of the warm plasma dispersion relation in an infinite homogeneous plasma. However, it does not mean that the observations rule out alternative possible generation mechanism of beads/rays, such as ionospheric feedback instability (IFI)/ionospheric Alfvén resonator (IAR) (Sakaguchi et al., 2009; Kataoka et al., 2011) that may occur in M-I coupling region closer to the Earth. If the authors have any special reason to rule out the possibility of IFI/IAR, they should provide evidence and discuss.

(3) The authors do not discuss the azimuthal propagation nature of auroral beads that may provide important clues for determining their driver. It should be considered whether or not the azimuthal propagation speed/direction of auroral beads can be interpreted in the context of kinetic Alfvén waves.

(4) What is the point of using 3-Hz (0.33 s) sampling data from the 557.7 nm (green-line) wavelength auroral emissions observed by the MOOSE imagers? The radiative lifetime of the green-line emission is 0.7 s (Vallance Jones, A., *Aurora*, 1974), longer than the sampling rate. If the authors would like to take advantage of the high sampling rate measurements, using the MOOSE data at 427.8 or 844.6 nm (which is a prompt emission compared to 557.7 nm) would be better for this sort of data analysis.

Reviewer #3 (Remarks to the Author):

Review of 'A direct diagnosis of the plasma waves responsible for the explosive energy release at substorm onset.

The manuscript presents results from a Fourier analysis of optical imagery at the onset of a geomagnetic substorm. The main thrust is that the variation in the spatial and temporal variation in luminosity is well matched by the expected dispersion of kinetic Alfvén waves in the equatorial plane. It is suggested that this observation can be used to distinguish (or in this case eliminate) candidate models for the substorm onset process. The manuscript reads well and makes its key points clear. The measurements of the dispersion characteristics at onset is an important new result that in my opinion should be published. However, there are several significant issues that need to be addressed to more correctly represent this work and to place it in context with that which has already been performed.

1. There are significant uncertainties in the measurements and subsequent interpretation along with rather broad assumptions that make the use of words such as 'perfect' (page 12 line 9 and top of page 13) and 'conclusive' (page 1 last line) perhaps too strong. Specifically:

- a) There is a broad spread and often multiple peaks (or are these harmonics) in the w measurements at each wavenumber (Figure 2a). From visual inspection it looks like the width at half maximum of the distribution in w at each k is larger than the peak value. I agree that there appears to be agreement but it is suggestive rather than 'conclusive'. Perhaps the authors can do some statistical analysis to include on the plot to quantify how good the agreement actually is.
 - b) The vertical extent of the emission at 557.7nm is comparable to the horizontal wavelengths in the luminosity considered. While this is not an issue for features directly above the camera it does introduce ambiguity away from the center of the camera field of view. This introduces uncertainty into what is actually being measured. Given that Figure 1b shows results compiled over 3 degrees in longitude this may be significant. If this is not significant then it needs to be demonstrated in the methods section.
 - c) The extrapolation of these measurements through the auroral acceleration region and into the equatorial plane along the geomagnetic field more than 10 R_E when the parallel wavelength of the wave is suggested to be 1.8 R_E is an 'untested' and perhaps unverifiable assumption. It might work like this but it needs to be made clear that this is a supposition and not really known. Perhaps an appropriate reference to simulation results could help here.
 - d) The features in the imagery do not look like plane waves. The gradients in the latitudinal direction are larger than in the assumed longitudinal direction of k . It may therefore not be valid to compare such features with frequencies and wavenumbers derived from a plane wave model. I think some qualification is needed on this point.
 - e) The lack of actual measurements of the conjugate plasma parameters in this equatorial plane. While the authors do consider the effect of variations in these, to claim a 'conclusive' result perhaps requires a near conjunction with a spacecraft in the equatorial plane.
- These are not damning criticisms, I think the analysis is basically correct. Each point above represents something that in my opinion was not well represented or justified in the text rather than wrong.

2. It is not clear that this work 'extends plasma theory into the high beta regime' (Abstract First page line 13 and elsewhere). As far as I can tell there is no new theoretical development here. If there is please specify. Numerical warm plasma dispersion solvers valid for the parameter range considered are readily available to the community [e.g. Ronnmark, 1982] and kinetic Alfvén wave dispersion results for plasma beta larger than 1 have previously been explored [Howes et al., 2006] and compared to in-situ observations [Chaston et al., 2009] without the requirement that $v_{\text{thermal}}/v_{\text{Alfvén}}$ be less than 10 as asserted on page 7 line 10. I have not seen observations reporting dispersion results at beta as large as 20 considered here but unless this introduces new physics it is not an 'extension of the plasma theory'. So, unless there is something new in the

section entitled 'Extension of current theory into the warm plasma regime' it should probably be removed along with claims of extending the plasma theory.

Ronnmark, K., WHAMP-waves in a homogeneous, anisotropic, multi-component plasmas, Rep. 179, Kiruna Geophys. Inst., Sweden, 1982.

Chaston, C. C., J. R. Johnson, M. Wilber, M. Acuna, M.L. Goldstein, and H. Reme, Kinetic Alfvén wave turbulence and transport through a reconnection diffusion region, *Phys. Rev. Lett.*, 102, 015001, 2009.

Gregory G. Howes, Steven C. Cowley, William Dorland, Gregory W. Hammett, Eliot Quataert, and Alexander A. Schekochihin, *Ap. J.*, 651:590–614, 2006.

Salem, C. S., G. G. Howes, D. Sundkvist, S. D. Bale, C. C. Chaston, C. H. K. Chen and F. S. Mozer, Identification of kinetic Alfvén wave turbulence in the solar wind (2012), *Ap. J.*, 745:L9, doi.org/10.1088/2041-8205/745/1/L9.

3. The use of the word 'direct' in the title here is inaccurate. Rather the temporal and spatial wave scales are assumed to be projections from the equatorial plane over $\sim 10 R_E$ along the field line down to the altitude of visible emission.

4. While this work is to my knowledge the first to use auroral imagery to identify Alfvén waves specifically at substorm onset, it is not the first use of optical observations to identify Alfvén wave dispersion as asserted in the abstract (line 11). See for example Semeter et al 2006; 2008 and Chaston et al. 2010.

Semeter, J., and E. M. Blixt (2006), Evidence for Alfvén wave dispersion identified in high-resolution auroral imagery, *Geophys. Res. Lett.*, 33, L13106, doi: 10.1029/2006GL026274.

Semeter, J., M. Zettergren, M. Diaz, and S. Mende (2008), Wave dispersion and the discrete aurora: New constraints derived from high-speed imagery, *J. Geophys. Res.*, 113, A12208, doi: 10.1029/2008JA013122.

Chaston, C. C., K. Seki, T. Sakanoi, K. Asamura, M. Hirahara, Motion of Aurorae, *Geophys., Res. Lett.*, 37, L08104, doi:10.1029/2009GL042117, 2010.

5. There is a several reports of in-situ observations of kinetic Alfvén waves in the equatorial plane and along the same field-lines as detailed in this study - some in fact at substorm onset conjunctive with auroral imaging [Hull et al., 2016; Ergun et al 2015; Chaston et al., 2012; Angelopoulos et al., 2002]. While it is appropriate to acknowledge these works their inclusion supports the plausibility of the interpretation given in the present manuscript.

Hull, A. J., C. C. Chaston, H. U. Frey, M. O. Fillingim, M. L. Goldstein, J. W. Bonnell, and F. S. Mozer (2016), The "Alfvénic surge" at substorm onset/expansion and the formation of "Inverted Vs": Cluster and IMAGE observations, *J. Geophys. Res. Space Physics*, 121, 3978–4004, doi: 10.1002/2015JA022000.

Ergun, R. E., K. A. Goodrich, J. E. Stawarz, L. Andersson, and V. Angelopoulos (2015), Large-amplitude electric fields associated with bursty bulk flow braking in the Earth's plasma sheet, *J. Geophys. Res. Space Physics*, 120, 1832–1844, doi:10.1002/2014JA020165.

Chaston, C. C., J. W. Bonnell, L. Clausen, and V. Angelopoulos (2012), Correction to "Energy transport by kinetic-scale electromagnetic waves in fast plasma sheet flows", *J. Geophys. Res.*, 117, A12205, doi:10.1029/2012JA018476.

Angelopoulos, V., J. A. Chapman, F. S. Mozer, J. D. Scudder, C. T. Russell, K. Tsuruda, T. Mukai,

T. J. Hughes, and K. Yumoto (2002), Plasma sheet electromagnetic power generation and its dissipation along auroral field lines, *J. Geophys. Res.*, 107(A8), 1181, doi:10.1029/2001JA900136.

6. Page 3 line 6, 7, 8: While I agree that the identification of a mode is important, can it really be said that 'the key physical attributes' of substorm onset are identified in this study. I would say instead that 'one of the key physical attributes is identified'.

7. Page 4 line 5: Has it really been shown in this manuscript that 'substorm aurora is caused by electron acceleration in shear Alfvén waves'?

8. Page 4 2nd paragraph, line 6. I think you need to indicate on the figure the feature to which you are referring.

9. Page 4 last line. There is a very long history concerning the distribution of auroral arc widths. It's a simple thing, but it would be useful to express this wavenumber as a scale or wavelength as done for the extrapolated results in the equatorial plane so the reader immediately knows the size of the features being considered.

Three reviews were received and we address each point-by-point below, the Reviewer's comments are in plain text, our responses are in bold and any additions to the text are quoted in bolded italics with line numbers of the revised manuscript.

Reviewer #1 (Remarks to the Author):

The onset is a major issue of auroral substorms/magneospheric substorms and has been hotly debated. The paper deals with one of the precursors called "beads" which was discovered some years ago. The paper consists of an excellent data set and a good insight into details of theoretical analysis of the waves associated with this phenomenon. Therefore, I am happy to recommend its publication in Nature.

We sincerely thank the Reviewer for their comments.

Reviewer #2 (Remarks to the Author):

This paper has attempted to construct the dispersion relation of the auroral structuring to diagnose the waves responsible for the explosive energy release from substorm onset, using high-resolution auroral imagery data. The authors have examined a particular case of auroral beads developing along an onset arc on 18 September 2012 and demonstrated that the azimuthal structuring of the arc is the manifestation of kinetic Alfvén waves driven in the high-beta regime of the near-Earth magnetotail, not any of previously proposed instabilities (such as, cross-field current instability, ballooning instability). This may have potentially important implications towards the understanding of how these features relate to magnetospheric activity and the importance and relevance to magnetospheric dynamics around substorm onset. However, I cannot recommend that this manuscript is published in the current form due to the lack of novel dataset/approach. Below are specific comments.

We are pleased to see that the Reviewer has identified that we have demonstrated that the azimuthal structuring of the substorm onset arc is the manifestation of kinetic Alfvén waves in the high-beta regime of the near-Earth tail. The combination of the first observational dispersion relation produced from auroral measurements and the link to theory is certainly a new result, and one which the Reviewer identifies as such. Below, we answer the specific comments of the Reviewer in order.

(1) The dataset and approach to determine the properties of the azimuthal structure of auroral beads are NOT novel, which is a major weakness of this study. Indeed, Sakaguchi et al. ("Azimuthal structures of ray auroras at the beginning of auroral substorms", *Geophys. Res. Lett.*, 2009) and Kataoka et al. ("Turbulent microstructures and formation of folds in auroral breakup arc", *J. Geophys. Res.*, 2011) have already estimated the growth rates and spatial structuring characteristics of onset arc based on similar high temporal/spatial resolution auroral image data and approach.

We agree with the Reviewer that the dataset used in our study is, in itself, not novel; high spatial measurements by, for example, Sakaguchi et al., 2009; Kataoka et al., 2011 of auroral beads have been presented at 29.97 Hz and 110 Hz, respectively. These studies were performed with white-light and prompt 845 nm wavelength emissions. In short, wavelength-resolved all-sky cameras have indeed been used before to study auroral beads.

Our approach, however, is certainly novel. We build upon previous work (e.g., Sakaguchi et al., 2009; Rae et al., 2010; Kataoka et al., 2011; Kalmoni et al., 2015) that has characterised the spatial scales and growth rates of the waves, but these determinations alone are insufficient to diagnose the type of plasma mode operating. We show that it is crucial to look at the temporal variation and spatial structure of the beads. It is only by looking at both structure and frequency of the undulations that we can diagnose the type of unstable magnetospheric plasma waves that are present in this instance. We use the entirety of these observed physical characteristics to construct the first observational dispersion relation from auroral data, and compare with theoretical expectations. It is the combination of these critical new observational results and theoretical calculations that is key to understanding what processes are responsible for creating auroral beads, the first signature of the substorm instability in the ionosphere.

Below, we briefly summarise the two manuscripts the Reviewer cites, to put our novel analysis in context.

In plasma physics, there are three key characteristics that define an instability and make up the dispersion relation; the temporal characteristics which can be split into real and imaginary components (Treumann and Baumjohann, 1997), and the spatial characteristics. Previous work only has only been able to determine either the spatial scales (e.g., Sakaguchi et al., 2009), or the spatial scales and imaginary component, or growth rate, of the instability (e.g., Rae et al., 2009; Kalmoni et al., 2015). Here, we are additionally able to determine the real component of the temporal wave characteristics and it is this quantity that allows us to construct the first observational dispersion relation of the substorm instability.

Kataoka et al. (2011) showed the temporal evolution of auroral intensities as a function of wavenumber on a minute-by-minute basis (Figure 5, Kataoka et al., 2011) for a region of perpendicular scales below 16 km. From this information, it would be possible to derive the wave growth rates. Additionally, no results describing the real part of the frequency corresponding to each spatial scale are presented. Kataoka et al.'s results provide some insight into the physics operating in the instability on minute timescales and for a limited set of spatial scales. This spatial and temporal region covers a small fraction of the dispersion relation that we have derived in Figure 2 (above $21.6 \times 10^{-6} \text{ m}^{-1}$, which corresponds to 16km spatial scales in the ionosphere), and at a single (60 second) temporal resolution. In contrast to this, our results clearly show that the physics operating at substorm onset occurs on significantly shorter timescales ($\leq 10\text{s}$) and larger spatial scales ($\geq 20\text{km}$).

Sakaguchi et al. (2009) also showed the temporal evolution of auroral features with different spatial scales, as discussed above. Sakaguchi et al. (2009) also estimate the growth in total auroral intensity as a function of time. However, the exponential increase in total auroral intensity should not be confused with the imaginary part of the temporal wave frequency, which is a function of wavenumber, as discussed by Kalmoni et al. (2015).

We would like to emphasise that this manuscript presents a determination of the both the real and imaginary components of the temporal frequencies as a function of wavenumber, which is used to construct the dispersion relation. This is new physics extracted from high cadence auroral imaging which allows us to pinpoint the substorm onset instability with additional information from current theory.

We have included the following citations in the manuscript:

Treumann and Baumjohann, 1997 (p5, L10-11 of the manuscript), Kataoka et al., 2011 (p3, L19 of the manuscript)

(2) The interpretation of the presented results remains largely speculative due to the lack of direct evidence from in situ measurements. The dispersion relation extracted from the aurora image data generally fits the frequency solutions expected from the theoretical approach of the warm plasma dispersion relation in an infinite homogenous plasma.

However, it does not mean that the observations rule out alternative possible generation mechanism of beads/rays, such as ionospheric feedback instability (IFI)/ionospheric Alfvén resonator (IAR) (Sakaguchi et al., 2009; Kataoka et al., 2011) that may occur in M-I coupling region closer to the Earth. If the authors have any special reason to rule out the possibility of IFI/IAR, they should provide evidence and discuss.

We agree with the Reviewer that there is a lack of direct evidence from in situ measurements, but we would like to reiterate that our new observational dispersion relation is backed up by extended theoretical calculations that show that shear Alfvén waves with short perpendicular scales can remarkably well reproduce the dispersion curve that we observe. Hence, we would not characterise our results as “speculative”. It is true that we cannot rule other instabilities out without in-situ observations, however, we would point out that the references discussed have the following characteristics which are not consistent with our findings.

Kataoka et al. (2011) states that the IAR has periods of ~seconds (Lysak, 1991). We show an extended dispersion relation in omega to demonstrate that there are no data points in the region up to, and including, second timescales which would be expected from interaction with the Ionospheric Alfvén Resonator

Sakaguchi et al. show that wavenumbers are concentrated at specific scales. The brightening aurora immediately cascades into smaller structures after a few seconds. Similar dynamics of dispersive Alfvén waves have been observed in simulations by Lysak and Song (2008). We see from Figure 1(c) (manuscript) characteristics of an inverse cascade where the dominant/characteristic wavenumbers decrease with time (e.g. Lui, 2008; Rae et al., 2010 Kalmoni et al., 2015). We can hence conclude that kinetic Alfvén waves generated in the near-Earth plasmasheet are the most likely mechanism with which to drive our observed wave signatures, backing up our theoretical results.

We have included the following paragraph on p6-7, L20-6.

“We note that the temporal cadence of the auroral imagers allows us to measure wave frequencies up to the Nyquist frequency of $\omega_r = 11 \text{ rads}^{-1}$. In Figure 2a we only present the dispersion relation up to $\omega_r = 0.8 \text{ rads}^{-1}$ as there are no datapoints at these higher frequencies. From the lack of occurrence at frequencies above 0.8 rads^{-1} we can rule out that the beads in this event are caused by the Ionospheric Alfvén Resonator which has periods of \sim seconds (Lysak, 1991) as previously suggested by Sakaguchi et al. (2009) and Kataoka et al. (2011). Moreover, Figure 1c shows an inverse cascade where the characteristic wavenumber of the beads decreases following substorm onset, in agreement with Lui et al., (2008); Rae et al., (2010); Kalmoni et al., (2015; 2017), which also supports the conclusion that the near-Earth magnetotail is the source of the instability.”

(3) The authors do not discuss the azimuthal propagation nature of auroral beads that may provide important clues for determining their driver. It should be considered whether or not the azimuthal propagation speed/direction of auroral beads can be interpreted in the context of kinetic Alfvén waves.

We thank the Reviewer for suggesting this independent verification of our results. When we studied the perpendicular propagation velocity in the ionosphere of the auroral beads, we find that their velocities are in the range 1.1-1.3 km/s (p4, L20), which corresponds to 20-23km/s in the magnetosphere using our mapping technique (p6 L16-19). From the solutions to the warm plasma dispersion relation presented in Figure 2, we find that the perpendicular phase velocity, $v_{ph} = \omega/k_{\perp}$, is in the same range above $k_{\perp} \sim 5 \times 10^{-6} \text{ m}^{-1}$ (see below Figure), with the dominant auroral bead scales being significantly higher at $k_{\perp} \sim 10 \times 10^{-6} \text{ m}^{-1}$. We hence include this Figure in the Results section, p11 and we thank the Reviewer for this additional evidence in support of our conclusion.

We have included the Figure (p10) and accompanying text (p9-10, L20-2) to describe this additional result.

“We can use the solutions of the full warm plasma dispersion relation to calculate a theoretical perpendicular velocity of the waves, given by $v_{\perp} = \frac{\omega_r}{k_{\perp}}$. Figure 3 shows the remarkable agreement between the phase velocities obtained from the solutions of the warm plasma dispersion relation for a wave with $\lambda_{\parallel} = 1.8 R_E$ and the bead along-arc propagation velocities determined from observations from $k_{\perp} \sim 5 \times 10^{-6} \text{ m}^{-1}$, with the dominant auroral bead scales being significantly higher at $k_{\perp} \sim 10 \times 10^{-6} \text{ m}^{-1}$.”

And on p10, L13-15

“Moreover, since our perpendicular phase velocity estimates are also constant, we believe that a choice of a constant parallel wavelength is appropriate.”

(4) What is the point of using 3-Hz (0.33 s) sampling data from the 557.7 nm (green-line) wavelength auroral emissions observed by the MOOSE imagers? The radiative lifetime of the green-line emission is 0.7 s (Vallance Jones, A., Aurora, 1974), longer than the sampling rate. If the authors would like to take advantage of the high sampling rate measurements, using the MOOSE data at 427.8 or 844.6 nm (which is a prompt emission compared to 557.7 nm) would be better for this sort of data analysis.

We agree with the Reviewer that there are more prompt emissions to be used from auroral measurements. However, we chose 557.7 nm for the following reasons: this emission is the brightest emission from the substorm onset process and is also prevalent before substorm onset, allowing us to track the substorm onset arc for a significantly longer period than other wavelengths. This is necessary as it allows us to determine wave periods which are relatively long in comparison to the analysis duration.

With regards to emission lifetimes, it is true that the lifetime of 557.7nm is 0.7s, but the production of 557.7nm is still prompt and allows us to see where the production is occurring; the 0.7s emission lifetime refers to the time that the emission takes to reduce by a factor of e. We would also point out that the periods of auroral beads are much longer than either the emission lifetime or the Nyquist period of the data (~10s compared to ~1s).

We thank the Reviewer for their constructive comments on our manuscript which we believe have greatly improved the context of our work and the evidence in support of our conclusions.

Reviewer #3 (Remarks to the Author):

Review of 'A direct diagnosis of the plasma waves responsible for the explosive energy release at substorm onset.

The manuscript presents results from a Fourier analysis of optical imagery at the onset of a geomagnetic substorm. The main thrust is that the variation in the spatial and temporal variation in luminosity is well matched by the expected dispersion of kinetic Alfvén waves in the equatorial plane. It is suggested that this observation can be used to distinguish (or in this case eliminate) candidate models for the substorm onset process. The manuscript reads well and makes its key points clear. The measurements of the dispersion characteristics at onset is an important new result that in my opinion should be published. However, there are several significant issues that need to be addressed to more correctly represent this work and to place it in context with that which has already been performed.

We are delighted to see that the Reviewer recommends publication of our important new results once we have satisfactorily addressed the Reviewers comments on context and representation.

1. There are significant uncertainties in the measurements and subsequent interpretation along with rather broad assumptions that make the use of words such as 'perfect' (page 12 line 9 and top of page 13) and 'conclusive' (page 1 last line) perhaps too strong. Specifically: **We agree with the Reviewer that we did not discuss the assumptions and uncertainties in our measurement and how these translate to our interpretation. We also agree regarding the choice of language used and have toned down these statements within the paper to accurately reflect our findings.**

a) There is a broad spread and often multiple peaks (or are these harmonics) in the w measurements at each wavenumber (Figure 2a). From visual inspection it looks like the width at half maximum of the distribution in w at each k is larger than the peak value. I agree that there appears to be agreement but it is suggestive rather than 'conclusive'. Perhaps the authors can do some statistical analysis to include on the plot to quantify how good the agreement actually is.

We have done as the Reviewer suggested in order to quantify how good our agreement is and we show this in two new Figures in the Methods (p20 and p23) section of our manuscript. We have taken the peak occurrence and Full Width Half Maximum (FWHM) at each k in order to investigate the broadness of our peaks. The figure below (Figure 7 in the manuscript) shows that the vast majority of the occurrences lie along the more visually identified dispersion relation we identified in our original manuscript, giving us increased confidence of our results.

We further compare the results from the peak and FWHM occurrences in Figure 7 with the dispersion relation results, which is shown below and in Figure 9 in the manuscript. We find that the chi squared value between the observed and theoretical results is 1.15, and is statistically significant to beyond the 99.99% confidence level (with 23 degrees of freedom), again giving us increased confidence that our results are indeed in strong agreement with theory.

We have added paragraphs on p20, L1-6 and p23, L1-7.

“This analysis results in the observational dispersion relation for the real frequency component, ω as a function of wavenumber. Figure 7 shows the observational dispersion relation, which is the same as shown in Figure 2a, with the peak occurrences for each wavenumber indicated by larger circles, and the Full Width Half Maximum of the peaks indicated by the smaller circles. Figure 7 shows that for the majority the peaks very clearly follow the line, with some exceptions particularly at higher wavenumbers.”

“The results from the theoretical dispersion relation have been indicated in Figure 2 for a parallel wavelength of $1.8 R_e \pm 10\%$. We use this theoretical calculation, together with the occurrence peaks from the dispersion relation (represented by the large circles in Figure 7) to perform a Chi-squared test between the observational and expected results. We find that the Chi squared value is 1.15, and is statistically significant to beyond the 99.99% level, indicating that the observations are very close to the expected values for a kinetic Alfvén wave with these characteristics. “

b) The vertical extent of the emission at 557.7nm is comparable to the horizontal wavelengths in the luminosity considered. While this is not an issue for features directly above the camera it does introduce ambiguity away from the center of the camera field of view. This introduces uncertainty into what is actually being measured. Given that Figure 1b shows results compiled over 3 degrees in longitude this may be significant. If this is not significant then it needs to be demonstrated in the methods section.

We agree with the Reviewer that elevation angle introduces a small amount uncertainty into the evolution of auroral bead spatial scales, which has been quantified in previous work to be small compared to the bead wavelength (e.g., Kalmoni et al., 2017). In this event, the extent of the arc is a maximum of 60 degree in elevation from zenith, which would introduce a maximum of 15-20 km on our ~ 100 km spatial scales. However, since we apply a windowing function to our measurements to perform the subsequent Fourier analysis, we essentially reduce the power in those larger elevation angles regions to zero, and the vast majority of our estimates of spatial scales are from the centre of our field-of-view.

We have included a statement to this effect on page 17, L2-13 to make this point clear.

“In producing the along-arc keogram, it is assumed that the 557.7 nm auroral emission is located at 110 km altitude and has very little vertical extent. This assumption does not affect the projection of the beads at zenith of the imager, but could have some effects if the assumption does not hold towards the edges of the auroral arc observed. In this event, the maximum extent of the arc away from zenith is 60° . This could introduce a maximum uncertainty of 15-20 km on our ~ 100 km bead spatial scales assuming that the 557.7 nm emission can extend between 100-120 km altitude. However, since we are applying a windowing function to our measurements to perform subsequent Fourier analysis, the error on these measurements is much smaller as the signature of the beads is weighted towards the observations towards the centre of the field of view.”

c) The extrapolation of these measurements through the auroral acceleration region and into the equatorial plane along the geomagnetic field more than 10 R_e when the parallel wavelength of the wave is suggested to be 1.8 R_e is an ‘untested’ and perhaps unverifiable

assumption. It might work like this but it needs to be made clear that this is a supposition and not really known. Perhaps an appropriate reference to simulation results could help here.

We agree and have included a paragraph on p10, L8-15.

“We reiterate that we have chosen a parallel wavelength of 1.8RE in order to match our observational results. Although the results of the warm plasma dispersion relation are relatively insensitive to the choice of k_{\parallel} , in the real magnetosphere, this wavelength may not be constant along the geomagnetic field line, particularly when travelling along the field into regions of increasing Alfvén speed (e.g. Watt and Rankin, 2009). Nevertheless, using a constant parallel wavelength, we find excellent agreement with our observational results. Moreover, since our perpendicular phase velocity estimates are also constant, we believe the choice of a constant parallel wavelength is appropriate.”

d) The features in the imagery do not look like plane waves. The gradients in the latitudinal direction are larger than in the assumed longitudinal direction of k . It may therefore not be valid to compare such features with frequencies and wavenumbers derived from a plane wave model. I think some qualification is needed on this point.

We agree with the Reviewer that auroral beads eventually wrap up into vortices and hence violate plane wave assumptions. However, the initiation of the auroral beads is more plane wave like (Figure 1a) up until around 09:25:00 UT, which is 75% of the interval studied (0922-0926 UT). In order to demonstrate this, we have performed the analysis on this shorter timescale 0921-0925 UT. The new truncated dispersion relation is shown below with the FWHM of the highest peaks for the entire analysis duration. Although there is a lot more power at low omega, which is to be expected as these frequencies are closer to our window length, this Figure shows that the peaks during the linear phase of the instability are still correctly determined during the more extended period we show in the manuscript.

We have added a note in the Methods section on the planar wave assumption but if the Reviewer felt it necessary to include details of this justification in this section, we would be more than happy to oblige in further reviews.

P21, L16

“Theoretical treatments are derived from a planar wave assumption.”

Figure 1 Dispersion relation from the data between 09:22:00-09:24:58 UT. FWHM are overplotted for the peak values in omega for the entire duration of the event (09:22:00-09:26:00)

e) The lack of actual measurements of the conjugate plasma parameters in this equatorial plane. While the authors do consider the effect of variations in these, to claim a ‘conclusive’ result perhaps requires a near conjunction with a spacecraft in the equatorial plane.
We agree that we should not use this language and have changed this throughout.

These are not damning criticisms, I think the analysis is basically correct. Each point above represents something that in my opinion was not well represented or justified in the text rather than wrong.

We thank the Reviewer for this comment.

2. It is not clear that this work ‘extends plasma theory into the high beta regime’ (Abstract First page line 13 and elsewhere). As far as I can tell there is no new theoretical development here. If there is please specify. Numerical warm plasma dispersion solvers valid for the parameter range considered are readily available to the community [e.g. Ronnmark, 1982] and kinetic Alfvén wave dispersion results for plasma beta larger than 1 have previously been explored [Howes et al., 2006] and compared to in-situ observations [Chaston et al., 2009] without the requirement that $v_{\text{thermal}}/v_{\text{Alfvén}}$ be less than 10 as asserted on page 7 line 10. I have not seen observations reporting dispersion results at beta as large as 20 considered here but unless this introduces new physics it is not an ‘extension of the plasma theory’. So, unless there is something new in the section entitled ‘Extension of current theory into the warm plasma regime’ it should probably be removed along with claims of extending the plasma theory.

We agree with the reviewer, it was not our intention to imply that we have made new theoretical developments, but rather to say that we use full kinetic warm plasma theory

and investigate (with evidence to constrain our choices) a higher beta regime than is normally considered.

We have revised these statements to describe that we apply full kinetic warm plasma theory and investigate (with evidence to constrain our choices) a higher beta regime than is normally considered.

We have revised this paragraph on p7, L19-23 to read:

“Previous work has been focussed in the low-mid beta regime below $v_{th}/v_A \sim 10$ (Lysak and Lotko, 1996; Howes et al., 2006), or compared to in-situ observations where no requirement on beta was imposed (e.g., Chaston et al., 2009). A beta regime of 20, however, is far outside the bounds of current comparisons of ultra-low frequency shear Alfvén waves.”

Ronnmark, K., WHAMP-waves in a homogeneous, anisotropic, multi-component plasmas, Rep. 179, Kiruna Geophys. Inst., Sweden, 1982.

Chaston, C. C., J. R. Johnson, M. Wilber, M. Acuna, M.L. Goldstein, and H. Reme, Kinetic Alfvén wave turbulence and transport through a reconnection diffusion region, Phys. Rev. Lett., 102, 015001, 2009.

Gregory G. Howes, Steven C. Cowley, William Dorland, Gregory W. Hammett, Eliot Quataert, and Alexander A. Schekochihin, Ap. J., 651:590–614, 2006.

Salem, C. S., G. G. Howes, D. Sundkvist, S. D. Bale, C. C. Chaston, C. H. K. Chen and F. S. Mozer, Identification of kinetic Alfvén wave turbulence in the solar wind (2012), Ap. J., 745:L9, doi.org/10.1088/2041-8205/745/1/L9.

We have included the Chaston and Howes references as per above.

3. The use of the word ‘direct’ in the title here is inaccurate. Rather the temporal and spatial wave scales are assumed to be projections from the equatorial plane over $\sim 10 R_e$ along the field line down to the altitude of visible emission.

We agree and have removed the word ‘direct’.

4. While this work is to my knowledge the first to use auroral imagery to identify Alfvén waves specifically at substorm onset, it is not the first use of optical observations to identify Alfvén wave dispersion as asserted in the abstract (line 11). See for example Semeter et al 2006; 2008 and Chaston et al. 2010.

Semeter, J., and E. M. Blixt (2006), Evidence for Alfvén wave dispersion identified in high-resolution auroral imagery, Geophys. Res. Lett., 33, L13106, doi: 10.1029/2006GL026274.

Semeter, J., M. Zettergren, M. Diaz, and S. Mende (2008), Wave dispersion and the discrete aurora: New constraints derived from high-speed imagery, J. Geophys. Res., 113, A12208, doi: 10.1029/2008JA013122.

Chaston, C. C., K. Seki, T. Sakanoi, K. Asamura, M. Hirahara, Motion of Aurorae, Geophys., Res. Lett., 37, L08104, doi:10.1029/2009GL042117, 2010.

We agree and have revised this sentence appropriately. We have included these citations on p3, L10-12.

“For example, optical observations of the aurora have shown the existence of dispersive scale Alfvén waves in the quiet-time ionosphere (e.g., Semeter and Blixt, 2006; Semeter et al., 2008; Chaston et al., 2010)”

5. There is a several reports of in-situ observations of kinetic Alfvén waves in the equatorial plane and along the same field-lines as detailed in this study - some in fact at substorm onset conjunctive with auroral imaging [Hull et al., 2016; Ergun et al 2015; Chaston et al., 2012; Angelopoulos et al., 2002]. While it is appropriate to acknowledge these works their inclusion supports the plausibility of the interpretation given in the present manuscript.

Hull, A. J., C. C. Chaston, H. U. Frey, M. O. Fillingim, M. L. Goldstein, J. W. Bonnell, and F. S. Mozer (2016), The “Alfvénic surge” at substorm onset/expansion and the formation of “Inverted Vs”: Cluster and IMAGE observations, J. Geophys. Res. Space Physics, 121, 3978–4004, doi: 10.1002/2015JA022000.

Ergun, R. E., K. A. Goodrich, J. E. Stawarz, L. Andersson, and V. Angelopoulos (2015), Large-amplitude electric fields associated with bursty bulk flow braking in the Earth’s plasma sheet, J. Geophys. Res. Space Physics, 120, 1832–1844, doi:10.1002/2014JA020165.

Chaston, C. C., J. W. Bonnell, L. Clausen, and V. Angelopoulos (2012), Correction to “Energy transport by kinetic-scale electromagnetic waves in fast plasma sheet flows”, J. Geophys. Res., 117, A12205, doi:10.1029/2012JA018476.

Angelopoulos, V., J. A. Chapman, F. S. Mozer, J. D. Scudder, C. T. Russell, K. Tsuruda, T. Mukai, T. J. Hughes, and K. Yumoto (2002), Plasma sheet electromagnetic power generation and its dissipation along auroral field lines, J. Geophys. Res., 107(A8), 1181, doi:10.1029/2001JA900136.

We agree and have included these highly appropriate references on p13, L15-18.

“Indeed, in-situ observations have provided evidence for kinetic Alfvén waves in the equatorial magnetotail in this region closely conjugate to auroral brightening (Angelopoulos et al., 2002; Chaston et al., 2012 ; Ergun et al 2015; Hull et al., 2016).”

6. Page 3 line 6, 7, 8: While I agree that the identification of a mode is important, can it really be said that ‘the key physical attributes’ of substorm onset are identified in this study. I would say instead that ‘one of the key physical attributes is identified’.

We agree and have changed the text accordingly on p2, L21.

7. Page 4 line 5: Has it really been shown in this manuscript that ‘substorm aurora is caused by electron acceleration in shear Alfvén waves’?

We have revised this on p4, L1-4, to read:

“substorm onset aurora is strongly associated with shear Alfvén waves of short perpendicular extent. These waves accelerate electrons in the magnetosphere (e.g. Watt

et al., 2010) suggesting that the shear Alfvén waves embedded in the auroral signature are not just modifying the aurora, but likely the cause of it."

8. Page 4 2nd paragraph, line 6. I think you need to indicate on the figure the feature to which you are referring.

We agree and have included an arrow and added it to the description on p4, L11 in Figure 1a as the Reviewer suggested.

9. Page 4 last line. There is a very long history concerning the distribution of auroral arc widths. It's a simple thing, but it would be useful to express this wavenumber as a scale or wavelength as done for the extrapolated results in the equatorial plane so the reader immediately knows the size of the features being considered.

We agree. This wavenumber corresponds to a 60km ionospheric wavelength and we have included this number in parentheses.

We thank the Reviewer for their constructive comments on our manuscript which we believe have greatly improved the context of our work, the accuracy of our descriptions and the evidence in support of our conclusions.

Reviewers' comments:

Reviewer #2 (Remarks to the Author):

I think the authors made adequate revisions on many issues, but I would like the authors to consider one more point. If auroral beads are a manifestation of kinetic Alfvén waves, field-aligned Alfvénic electron fluxes are expected to be responsible for the bright emissions of the beads. However, Motoba and Hirahara [GRL, 2016] questioned this kind of explanation based on the fact that monoenergetic electron fluxes at larger pitch angles ($> 30^\circ$) contribute primarily to the bright emissions of the structuring arc, rather than field-aligned electron fluxes. Some discussion on this point is needed.

Motoba, T., and M. Hirahara (2016), High-resolution auroral acceleration signatures within a highly dynamic onset arc, *Geophys. Res. Lett.*, 43, doi:10.1002/2015GL067580.

Reviewer #3 (Remarks to the Author):

The authors have adequately addressed the comments from my previous review. There are however a couple of outstanding issues.

1. The reference I previously supplied by Hull et al 2016 which describes kinetic Alfvén waves at sub-storm onset from direct in-situ measurements. Here's the first line from the abstract of this work:

'From multipoint, in situ observations and imaging, we reveal the injection-powered, Alfvénic nature of auroral acceleration during onset and expansion of a substorm' ...which then goes on to describe dispersive Alfvén waves (i.e kinetic Alfvén waves)

The relevance of this work in my opinion requires more than just the addition of:

'evidence for kinetic Alfvén waves in the equatorial magnetotail in this region closely conjugate to auroral brightening'

Rather it could be explicitly stated that 'kinetic Alfvén waves at substorm onset have previously been identified from in-situ measurements (Hull et al., 2016)'

In my opinion a statement of this nature actually makes the assertions of the manuscript presently under review more plausible.

2. I should have raised this in my last review. However, the phase speed of the waves across the magnetic field in the equatorial plane is perhaps less than the flow speed of plasmas injected into this region of space during active times (e.g substorm onset. I believe the analysis technique assumes an equatorial background plasma supporting the wave that is at rest. It is therefore necessary to compare the derived phase speed of the wave to expected flow speeds to make sure that the apparent phase speed is not primarily a consequence of Doppler shift - structures embedded in the moving plasma for example. The geometry (i.e the orientation of k and expected v_{flow}) may be helpful here.

Reviewers' comments

We thank both Reviewers for their comments, which have added important final discussion of our results in relation to previous work. We provide a point-by-point response to both Reviewer 2 and Reviewer 3, each Reviewer comment is in plain text, our replies are in bold and any material added to the article in quoted bolded italics.

Reviewer #2 (Remarks to the Author)

I think the authors made adequate revisions on many issues, but I would like the authors to consider one more point. If auroral beads are a manifestation of kinetic Alfvén waves, field-aligned Alfvénic electron fluxes are expected to be responsible for the bright emissions of the beads. However, Motoba and Hirahara [GRL, 2016] questioned this kind of explanation based on the fact that monoenergetic electron fluxes at larger pitch angles ($> 30^\circ$) contribute primarily to the bright emissions of the structuring arc, rather than field-aligned electron fluxes. Some discussion on this point is needed.

Motoba, T., and M. Hirahara (2016), High-resolution auroral acceleration signatures within a highly dynamic onset arc, *Geophys. Res. Lett.*, 43, doi:10.1002/2015GL067580.

Motoba and Hirahara (2016) do question this explanation based upon the fact that there is more electron energy flux at larger pitch angles than is field-aligned. However, previous evidence from FAST spacecraft measurements and kinetic simulations of Alfvénic acceleration (Watt et al., 2006) indicates that acceleration due to large Alfvén waves can result in monoenergetic populations of electrons with a wide range of pitch angles (Figure 2, Watt et al., 2006). This is evidence that the electrons were accelerated at altitudes far above the spacecraft measurement, as confirmed by the simulations in Watt and Rankin (2009). We have added a discussion of this, as requested on page 14 line 20 to page 15 line 5 of the revised manuscript.

“Note that we have no low-altitude evidence of the type of electron acceleration mechanism that causes this type of aurora. Motoba and Hirahara (2016) showed evidence from low-altitude satellite measurements where electrons at a range of pitch angles are observed at the same time as the onset arc. previous evidence from FAST spacecraft measurements and kinetic simulations of Alfvénic acceleration (Watt et al., 2006) indicates that acceleration due to large Alfvén waves can result in monoenergetic populations of electrons with a wide range of pitch angles (Figure 2, Watt et al., 2006) and confirmed by the simulations of Watt and Rankin (2009). Both the in-situ and new ground-based results presented in this paper point to an Alfvén wave instability operating at substorm onset and providing an acceleration mechanism for the onset aurora.”

References

Watt, C. E. J., R. Rankin, I. J. Rae, and D. M. Wright (2006), Inertial Alfvén waves and acceleration of electrons in nonuniform magnetic fields, *Geophys. Res. Lett.*, 33, L02106, doi:10.1029/2005GL024779.

Watt, C. E. J., and R. Rankin (2009), Electron trapping in shear Alfvén waves that power the aurora, *Phys. Rev. Lett.*, 102(4), 045002.

Reviewer #3 (Remarks to the Author)

The authors have adequately addressed the comments from my previous review. There are however a couple of outstanding issues.

1. The reference I previously supplied by Hull et al 2016 which describes kinetic Alfvén waves at substorm onset from direct in-situ measurements. Here's the first line from the abstract of this work: 'From multipoint, in situ observations and imaging, we reveal the injection-powered, Alfvénic nature of auroral acceleration during onset and expansion of a substorm'which then goes on to describe dispersive Alfvén waves (i.e kinetic Alfvén waves)

The relevance of this work in my opinion requires more than just the addition of:

'evidence for kinetic Alfvén waves in the equatorial magnetotail in this region closely conjugate to auroral brightening'

Rather it could be explicitly stated that 'kinetic Alfvén waves at substorm onset have previously been identified from in-situ measurements (Hull et al., 2016)'

In my opinion a statement of this nature actually makes the assertions of the manuscript presently under review more plausible.

We agree and have revised this statement accordingly.

“Indeed, kinetic Alfvén waves at substorm onset have previously been identified from in-situ measurements (Hull et al., 2016) and kinetic Alfvén waves have been observed in the equatorial magnetotail in this region closely conjugate to auroral brightenings (Angelopoulos et al., 2002; Chaston et al., 2012 ; Ergun et al 2015).”

2. I should have raised this in my last review. However, the phase speed of the waves across the magnetic field in the equatorial plane is perhaps less than the flow speed of plasmas injected into this region of space during active times (e.g substorm onset. I believe the analysis technique assumes an equatorial background plasma supporting the wave that is at rest. It is therefore necessary to compare the derived phase speed of the wave to expected flow speeds to make sure that the apparent phase speed is not primarily a consequence of Doppler shift - structures embedded in the moving plasma for example. The geometry (i.e the orientation of k and expected v_{flow}) may be helpful here.

The Reviewer is correct that our theoretical calculations assume a plasma at rest. For our observational analysis technique, we should also have emphasized the following point: our observational analysis window follows the latitudinal centroid of the substorm onset arc and therefore moves poleward during the interval as the substorm onset arc also moves poleward. This accounts for any apparent phase motion arising from plasma injected into the instability region and/or dipolarisation of the nightside field in the radial direction. Hence, we are confident that our analysis will obtain the true azimuthal phase speed of the auroral features.

We have added the following statement page 10 line 22 to page 11 line 4.

“Note that our observational analysis window follows the latitudinal centroid of the substorm onset arc (see Methods) and therefore moves poleward during the interval as the substorm onset arc also moves poleward. This accounts for any apparent phase motion arising from plasma injected into the instability region and/or dipolarisation of the nightside field in the radial direction. Hence, we are confident that our analysis will obtain the true azimuthal phase velocity of the auroral features.”